# Can a Digital Application for Animal Welfare Self-Assessments by Farmers Help Improve the Welfare of Free-Range and Organic Pigs?

**DOI:** 10.3390/ani14233374

**Published:** 2024-11-23

**Authors:** Evelien A. M. Graat, Charlotte Vanden Hole, T. Bas Rodenburg, Mona F. Giersberg, Frank A. M. Tuyttens

**Affiliations:** 1Animal Sciences Unit, ILVO, 9090 Melle, Belgium; evelien.graat@ilvo.vlaanderen.be (E.A.M.G.); charlotte.vandenhole@ilvo.vlaanderen.be (C.V.H.); 2Department of Veterinary and Biosciences, Faculty of Veterinary Medicine, Ghent University, 9820 Merelbeke, Belgium; 3Faculty of Veterinary Medicine, Utrecht University, 3584 CL Utrecht, The Netherlands; t.b.rodenburg@uu.nl (T.B.R.); m.f.giersberg@uu.nl (M.F.G.)

**Keywords:** pigs, animal welfare, free-range, organic, welfare self-assessment, smartphone application, benchmarking

## Abstract

In light of the European Green Deal, the EU has agreed that it is necessary to increase organic farming and improve the welfare of farm animals. This will likely lead to an increase in organic farm animals, including pigs. While organic farms are considered to be more animal friendly, the animals still suffer from certain welfare problems that need to be monitored and addressed. For this reason, we developed the PIGLOW app, which helps free-range and organic pig farmers assess the welfare of their own pigs. Farmers receive automated feedback on the results and can compare their welfare scores to those of other app users (benchmarking). A group of 12 farmers was followed for two years while they regularly used the PIGLOW app to observe their pigs. A researcher visited each farm at the beginning and end of the study to determine whether animal welfare had improved. While no clear improvement was seen for the majority of assessed welfare aspects, the PIGLOW app seemed to have helped farmers improve some specific aspects of welfare. Farmers found the app easy to use and found many of its elements useful. However, they only noticed small effects of the use of the app on their farms. Some farmers suggested that the PIGLOW app could be useful for new farmers who have less experience with animal welfare.

## 1. Introduction

In 2021, the European parliament adopted a resolution on the “Farm to Fork” strategy as part of the European Green deal [1]. The communication on this strategy mentioned an “urgent need” to increase organic farming and improve animal welfare in the EU [2]. To help reach the first goal, a set of actions was developed to increase the percentage of EU agricultural land that is used for organic farming to at least 25% by 2030 (compared to 9.1% in 2020) [3]. While this plan does not include specific targets for animal production, it is likely that organic livestock production, including that of pigs, will increase in the coming years. Thus, to reach the second goal of improving animal welfare, it is important to also look at solutions to animal welfare problems that are specific to organic—and perhaps other outdoor—farming systems. In 2021, approximately 1% of pigs in the EU were raised organically [4]. The share of free-range pigs is somewhat larger, with a median of 8% in the 17 member states for which data were available [5]. Organic and free-range systems are often assumed to uphold higher animal welfare standards than conventional farms [6]. While these systems do indeed have advantages for animal welfare, the pigs on these farms still experience welfare issues [7]. An important advantage is that outdoor access and increased space allowance promote the expression of natural behaviours, such as rooting and foraging [8,9]. However, outdoor access also increases the risk of parasitic infections and makes pigs more vulnerable to adverse weather conditions [10,11,12]. Extreme temperatures, wind, or precipitation can lead to increased restlessness, irritation, and aggression, but also to sunburn, injuries, or a reduced growth rate. In addition, while welfare issues such as tail biting, lameness, and diarrhoea generally seem to be more common in conventional systems, they are present in outdoor systems as well [6,7,12,13,14,15]. An added difficulty for upholding high animal welfare standards in outdoor systems is that the more extensive outdoor environment makes it more challenging for farmers to monitor the animals [12,15]. This could decrease the likelihood that welfare problems are recognised at an early stage.

One possible way to address these welfare issues and help farmers monitor their pigs more closely could be to provide farmers with a welfare self-assessment tool specifically tailored to outdoor production systems. This was the goal of the PIGLOW app [16], which was developed within the PPILOW project [17] by scientists and veterinarians in co-creation with stakeholders. The PIGLOW app includes welfare assessments (WAs) with animal-based welfare indicators (WIs) for sows (pregnant and farrowing) and fattening pigs (growers and finishers) that were designed to be completed by farmers. There is also an assessment for the depopulation process for which the health and behaviour of the pigs is observed shortly before transport to the slaughterhouse. The WIs and observation methods were specifically chosen to be suitable for various types of outdoor pig production systems. After completing an assessment, farmers receive a report with results that includes a score for each WI that was assessed. For each WI, automated feedback in the form of risk factors for related welfare problems is provided. In addition, benchmarking information is available that allows farmers to anonymously compare their scores per WI with other app users. The app recommends that farmers discuss their results with a veterinarian or other expert who can help them determine whether it is necessary to take action and come up with a plan that suits that specific farm. PIGLOW was first developed in English and subsequently translated into the eight additional native languages of project partners (Danish, Dutch, French, German, Italian, Finnish, Norwegian, and Romanian). This allows many farmers to use the app in their own language.

It has been shown in the past that stimulating farmers to think more actively about animal welfare problems can lead to an improvement of animal welfare [18,19]. Similarly, providing farmers with benchmarking information and promoting discussions on welfare issues among farmers have also been proven to motivate farmers to make welfare-related changes on their farms [20,21,22,23]. Furthermore, in a study on animal welfare self-assessments in which benchmarking information was not included, many farmers mentioned that they were unsure how to evaluate their results and how to know whether their results were good or bad [24]. In light of these research results, it was thought that periodic use of the PIGLOW app could help farmers improve the welfare of their pigs. In addition, using the PIGLOW app regularly could sensitize farmers to early signs of welfare problems, which could help them take action earlier and thereby avoid larger issues in the future. To find out whether regular use of the PIGLOW app does indeed help organic and free-range pig farmers improve the welfare of their pigs, a two-year longitudinal study was conducted. In this study, a group of farmers was asked to periodically perform WAs for their finisher pigs with the PIGLOW app.

## 2. Materials and Methods

### 2.1. Farms

Thirteen farmers with free-range or organic finisher pigs in Belgium and The Netherlands were initially recruited in 2020 and 2021 to participate in the longitudinal study (Table 1). Only farmers whose main profession was raising free-range or organic livestock were considered for the study. One farmer was excluded from the study after approximately one year because he had not yet completed any WAs with the PIGLOW app on his own, despite having received several reminders. All farmers read and signed an informed consent form before the start of the study. They thereby agreed to perform the required number of WAs with the app, fill out surveys, and let researchers collect additional welfare data on their farms. Many different channels were used to recruit farmers willing to participate in the study. Firstly, members of the Belgian National Practitioner Group were approached. This was a group made up of farmers and other stakeholders who were involved in the PPILOW project to give their opinion on the research being conducted. Subsequently, colleagues from PPILOW partner organisations, other research institutes, and farmers’ organisations in Belgium and The Netherlands were asked for their help to approach farmers they worked with. With their help, calls for participation were also distributed through several newsletters and other media channels aimed at (organic) farmers. Finally, existing databases of organic farmers and farm stores were consulted and extensive Google searches were conducted to find farmers who fit the scope of the project. Farmers were contacted by phone or email, either by us or by an intermediate party who knew them, and received an explanation of the study, its requirements, and the possible advantages of participation.

### 2.2. Farm Visits

At the beginning of the longitudinal study, each of the participating farms was visited by a researcher who was experienced with the PIGLOW app. During these initial visits, which took place between November 2020 and August 2021, the farmers performed the first WA for finisher pigs with the PIGLOW app in the presence of the researcher (Table 2). The farmers were able to ask the researcher questions if any details of the app were not clear and received a short explanation on how to view and interpret the results of the WAs. However, the researcher did not assist the farmers with answering questions posed in the WAs. During the same visit, the researcher conducted a more detailed WA of the pigs on the farm (Table 3). The data of that assessment served as the baseline animal welfare status of the farm before use of the PIGLOW app. The detailed WA included 13 individual-level WIs and 7 group-level WIs. Individual WIs were scored either by answering a yes/no question for each pig or on a scale from 0 (no welfare problem) to 100 (worst possible welfare problem). Group-level WIs were either scored as the percentage of animals in a group that was positive for that WI or a score from 0–2 or 0–3 was assigned, with 0 indicating no welfare problem. If available on the farm, at least 30 individual pigs and at least 5 groups were observed for each assessment. On some farms, it was not possible to achieve this number even though enough pigs were present on the farm. This had varying reasons, such as pigs being too nervous to be approached closely, pigs that were invisible in extensive outdoor enclosures with vegetation, and small indoor enclosures that could not safely be entered by humans. Where possible, pigs of various ages within the category of finisher pigs and pigs from multiple pens were included in the assessment. Pens in different areas of the building/farm were chosen and approximately equal numbers of pigs from each of the included pens were chosen for individual observations. Assessments were conducted indoors, outdoors, or both, depending on the type of housing and on where the pigs were at the time of the assessment. At the end of the two-year study, each farm was visited by a researcher once more. During this visit, the farmers performed the final WA with the PIGLOW app in the presence of the researcher. In addition, the researcher repeated the detailed WA from the beginning of the study to determine the final welfare status of the pigs on the farm. Details on the number of individual pigs and groups of pigs included in each of the detailed WAs are listed in Table 4. Differences among farms in the observed number of pigs and groups can be explained by differences in the sizes of the farms. All detailed WAs in Belgium were completed by the same researcher, while three additional researchers were involved in the completion of the detailed WAs in The Netherlands due to COVID-19 restrictions. All researchers were experienced in the performance of animal welfare assessments for pigs. After detailed instructions from the lead researcher (EG), researchers in Belgium and The Netherlands held separate training sessions at their own facilities in which the protocol for the detailed WA was tested.

### 2.3. Perodic Welfare Assessments with the PIGLOW App

The farmers were asked to perform at least six WAs for finisher pigs with the PIGLOW app, approximately equally divided across the study period. This included the first and last WAs performed during the visits from a researcher and four additional WAs. Each farmer received a reminder in the form of an email or a phone call when the moment to perform their next WA was coming up. Researchers kept track of whether each WA was performed by each farmer, but the data of these assessments were not used to analyse the welfare statuses of the farms. Seven farmers did indeed perform at least six WAs during the study, while the remaining five farmers performed five WAs.

### 2.4. Surveys

The farmers filled out two surveys during the study: one before using the PIGLOW app for the first time and one shortly after the last WA of the study had been completed. In the first survey, farmers were asked to rate the importance of 16 aspects of welfare that were included in the PIGLOW app, either in the assessments themselves or in the automated feedback on the results. They also rated the performance of their own farm on the same 16 welfare aspects. The final survey asked whether their opinions on the importance of these welfare aspects or the welfare of their pigs had changed due to the use of the PIGLOW app. The farmers were also asked to rate the user-friendliness and usefulness of various elements of the app. These included elements of the WAs themselves, such as photo material of welfare problems and text boxes with additional explanations for questions, and elements of the results, such as benchmarking information and several visual representations of the welfare scores. Answers to the majority of quantitative questions were given as a rating from 1 (worst) to 7 (best), with one question for which ratings from 1 to 10 could be given. The exact meaning of each rating differed per question (Table 5). For the questions on user-friendliness and usefulness, it was stated that farmers were allowed to give no response if they did not feel that they had used an element enough to form an opinion. The number of scores that was given therefore gave information about how many farmers had used each element of the app. The final survey also included several open questions in which farmers were asked to add remarks to explain their ratings. One farmer did not fill in the final survey. Therefore, data of 11 farmers were analysed for both surveys. The surveys were designed in the LimeSurvey online survey system (version 6.5.17+240715). They were first written in English and subsequently translated into Dutch and French by native speakers. Farmers received an email with a link and a personal password to open the survey. While this personal code pseudonymised the survey responses inside the database, the personal code of each farmer was known to one researcher, allowing them to verify which farmers had already responded to the surveys. Additionally, this made it possible to link the survey answers of individual farmers to the welfare data from their farms.

### 2.5. Analysis

To determine whether pig welfare on the farms had changed, the scores for the 19 WI assessed in the detailed WAs at the beginning and end were compared. For the WIs “Shivering” and “Too small”, the percentages of observed pigs that were scored as positive for that indicator were compared. For all other individual level WIs (Panting, Bad general state, Hernia, Laboured breathing, Covered with faeces, Skin wounds, Scratches, Ear lesions, Tail lesions, Skin irritation, Lameness) the values were the mean scores between 0 and 100 for all observed pigs. For the group-level WIs “Huddling”, “Fear of humans”, “Liquid faeces”, “Coughing”, and “Sneezing”, the values were the mean scores between 0 and 2 for all groups. Finally, for “Enrichment use”, the value was the percentage of animals that was scored as positive for this indicator during a behavioural scan of each group. Mean values per WI per farm were used for comparison. Because the data did not meet the assumptions for statistical models, descriptive statistics were used to compare animal welfare at the beginning and end. As the data were not normally distributed, the median value per WI for all 12 farms combined at the beginning (median B) and end (median E) was used. In addition, the number of farms that scored higher than 0 (imperfect score) for each WI and the number of farms for which the scores per WI had improved or worsened during the study were counted. An aspect of welfare was considered to have improved during the study only when the median score for the WI had clearly improved at the end and when there were at least two more farms with an improved score during the study than with a worsened score. A clear improvement of the median was defined as an improvement of at least 20%, with the additional criterion for WIs assessed between 0 and 100 being that the absolute improvement of the median should be at least 0.5 point. A negative change in a WI of these same proportions was considered as a deterioration of that welfare aspect. The number of scores higher than 0 reflected the potential for improvement, as perfect scores could not be improved any further. Survey responses from 11 farmers were also analysed with descriptive statistics. For all quantitative questions with numerical answers, the mean ratings given by the farmers and the related standard deviations were compared. For open questions, comments from farmers were manually selected to be part of the results. Comments were translated into English from the farmers’ native languages, Dutch and French. While not all comments for each question were included, care was taken to make a fair selection of comments that reflected all of the farmers’ opinions. As part of the results, the total number of comments that was made for each question was listed, and the number of comments reflecting a similar opinion as each of the chosen comments was listed behind them. Microsoft Excel 2016 was used for all descriptive statistics, and graphs were made in Microsoft Excel and Rstudio version 2023.12.1+402.

## 3. Results

### 3.1. Detailed Welfare Assessments

For 9 of the 19 WI that were assessed (Bad general state, Panting, Shivering, Hernia, Laboured breathing, Skin irritation, Huddling, Liquid faeces, Sneezing), the median score of the 12 farms was 0 at the beginning of the study (Table 6). This indicates that more than half of the farms had a perfect score for those WIs before using the PIGLOW app. The median for eight of those WI was still 0 at the end of the study, but the median for “Hernia” increased slightly (median E = 0.41). However, as the increase in the median was small and the number of farms with improved and worsened scores did not differ considerably, the data did not support a deterioration of this aspect.

For 9 of the 19 WIs (Too Small, Panting, Shivering, Hernia, Laboured breathing, Skin irritation, Huddling, Liquid faeces, Coughing), the number of farms for which the score improved was the same or differed by only one digit from the number of farms for which the score worsened during the study. Thus, for these indicators, no clear pattern of change across farms was supported.

For eight WIs, medians B and E differed from one another, and there was a pattern of scores on the farms predominantly changing in the same direction between the beginning and end of the study. For one of these, namely, “Fear of humans”, the median score was higher at the end vs. the start of the study (median B = 0.21; median E = 0.40), and the scores worsened for the majority of the farms. However, the difference between median B and E was not large enough to support a clear difference between the two moments. For four of the eight WIs, namely, “Skin wounds” (median B = 0.23; median E = 0.21), “Ear lesions” (median B = 0.31; median E = 0.13), “Tail lesions” (median B = 0.07; median E = 0), and “Enrichment use” (median B = 20.15; median E = 19.18), the improvement in the median was not large enough to meet the criteria for a considerable change. Therefore, the data did not support a clear change in these aspects of welfare. For the final three WIs, “Covered with faeces” (median B = 4.40; median E = 2.01; 6 improved scores), “Scratches” (median B = 3.20; median E = 1.60; 8 improved scores), and “Lameness” (median B = 1.63; median E = 0; 8 improved scores), there was a larger improvement in the median, and there were decidedly more farms with improved than worsened scores (Figure 1). This suggests an improvement of these WIs on the participating farms during the time of the study.

### 3.2. Surveys

#### 3.2.1. Opinions on Animal Welfare

At the beginning of the study, the farmers rated the importance of all 16 welfare aspects between 5.58 and 7 on a scale from 1 (not important at all) to 7 (very important) (Figure 2), indicating that all aspects were judged to be quite important. “Water availability” (7.00 ± 0), “Food availability” (6.91 ± 0.30), “Expressing positive behaviour” (6.55 ± 0.82), and “Enough space” (6.36 ± 1.03) received the highest mean ratings, while the lowest ratings were given for the importance of “Thermal comfort” (5.45 ± 1.13), “Hygiene” (5.55 ± 1.13), “Enrichment use” (5.73 ± 1.27), and “Floor type” (5.73 ± 1.19). When rating their own performance on these aspects of welfare on a scale from 1 (very badly) to 7 (very well), the farmers estimated that their farms performed best on “Water availability” (6.82 ± 0.40), “Food availability” (6.73 ± 0.47), “Expressing positive behaviour” (6.45 ± 0.82), and “Enough space” (6.36 ± 0.92) (Figure 3). Their estimated performance on the aspects of “Absence of lameness” (5.18 ± 1.17), “Hygiene” (5.27 ± 1.56), “Thermal comfort” (5.45 ± 1.29), and “Floor type” (5.55 ± 1.37) received the lowest scores. All mean ratings were between 5.45 and 7.00, indicating that the farmers thought their farms performed quite well.

#### 3.2.2. Perceived Effect of the PIGLOW App on Animal Welfare

When asked whether the use of the PIGLOW app had changed how important certain aspects of animal welfare were to them, the mean score of 11 farmers on a scale from 1 (not at all) to 7 (very much) was 4.00 (SD = 2.00), indicating a medium-sized effect. Eight farmers elaborated on their answer with a comment. Their scores for the question above are listed behind their comments:“If you assess the pigs with the app, you do look more closely.” (6) (1 similar comment)“The assessment of lesions, scratches, etc. on the bodies of pigs has evolved positively in my opinion. I pay more attention to it.” (5).“Aggression of pigs” (6)“Because we muck out the enclosures every day, we have a good image of the welfare of the animals. I don’t think we treat the animals differently.” (4) (2 similar comments)“Our whole farm already revolves around obtaining the best possible animal welfare, it’s our main goal.” (1)

These quotes indicate that the PIGLOW app helped some farmers in paying closer attention to their pigs or to certain specific welfare aspects, but did not have a large effect on the opinions of those who were already focused on animal welfare before using the app.

The farmers’ opinions on the effects of the use of the PIGLOW app on their own performance for each of 16 welfare aspects are shown in Figure 4. They were asked to rate the effect on a scale from 1 (severely deteriorated) to 7 (greatly improved). As a negative effect of the use of the app was possible, this question differed from others in the survey in the respect that ratings from 1 to 3 represented a negative effect and a rating of 4 represented no effect. The mean ratings for the 16 welfare aspects varied between 3.91 and 4.91. Thus, on average, farmers noticed no effect or small improvements for each of these welfare aspects on their farms. The lowest mean scores were given to “Food availability” (3.91 ± 1.04), “Water availability” (4.00 ± 0.77), “Feed (structure)” (4.00 ± 1.18), and “Feed (vitamins and minerals)” (4.00 ± 0.18), whereas the highest mean scores were given to “Reaction to humans” (4.91 ± 1.14), “Expressing positive behaviour” (4.91 ± 1.14), “Enrichment use” (4.82 ± 1.08), and “Absence of wounds/lesions” (4.82 ± 0.98). When asked whether, overall, they had noticed a positive impact of the use of the PIGLOW app on the welfare of their pigs, farmers gave a mean score of 3.82 (SD = 1.60) on a scale from 1 (not at all) to 7 (absolutely). As any rating higher than 1 indicated a positive effect, this score indicates that farmers perceived a medium-sized positive effect of the PIGLOW app on their animals’ welfare. The large standard deviation indicates that there were relatively large differences among farmers. Four of the farmers explained their answers further:
“More reflection on pig behaviour” (5)“It has made me look at the animals slightly differently, but we were already very focused on it” (2)“Using the application allows you to evaluate and therefore can suggest improving something based on the questions you are facing. I don’t know which breeder would change their breeding conditions after using the application.” (5)“We regularly have young people walking around here, and I find that the app has added value for them. Now they know what they should look at.” (4)

These comments indicate that, while some farmers feel the PIGLOW app did not have a large influence, for others, it changed how they look at certain aspects of animal welfare. The third quote offers the suggestion that the PIGLOW app could be useful for people who are new to assessing animal welfare or to pig farming in general.

#### 3.2.3. Opinions on the PIGLOW App

When asked to rate the PIGLOW app in general between 1 (worst) and 10 (best), farmers gave a mean rating of 8.09 (SD = 1.38). This seems to indicate that the farmers considered PIGLOW to be a good app.

The farmers’ ratings for how easy they found it to use various elements of the PIGLOW app on a scale from 1 (very difficult) to 7 (very easy) are shown in Figure 5. The highest mean score was given for “General use of the app” (6.73 ± 0.47), indicating that farmers found the app very easy to use. Mean ratings for individual elements of the app ranged from 6.13 to 6.55, which shows that all elements were judged to be very user-friendly. The elements that were part of the WA themselves were rated by all farmers, but elements of the results were only rated by seven to nine of the eleven farmers who filled out the survey. This likely reflects that some farmers did not use all elements of the results.

The mean ratings for the usefulness of the elements of the PIGLOW app ranged between 5.63 and 6.70 on a scale from 1 (not useful at all) to 7 (very useful), indicating that on average, the farmers found all of the aspects useful (Figure 6). Farmers found “Additional information for questions” (6.70 ± 0.48) the most useful, while the usefulness of “Photo material for questions” (5.62 ± 1.51) received the lowest mean rating. Not all elements were rated by all farmers, with the number of ratings per element varying between 6 and 11.

Finally, the farmers were asked to give suggestions for possible improvements to the app. The following suggestions and comments were made in response to this question:“None, it’s very good like this” (9)“I would maybe give some more practical tips in the app, or examples from real farms” (9)“More depth and better feedback on the results” (7)“Sometimes a little bit too precise” (9)

The suggestions show that farmers’ opinions on how the PIGLOW app should be improved differed from one another.

## 4. Discussion

### 4.1. The Effect of the PIGLOW App on Animal Welfare

The main goal of this study was to test whether repeated use of the PIGLOW app helps farmers improve the welfare of their pigs. A limitation of the study is that, unfortunately, the sample size was smaller than foreseen. This was largely due to the COVID-19 pandemic, which made it more challenging to recruit and visit farmers. The difficulties with recruiting farmers are also the reason for the absence of a control group. Because the group of farmers that was willing to participate fully in the study was too small to randomly assign half of them to the control group, the control group would have had to be made up of farmers who were not willing to use the PIGLOW app, but were willing to let researchers assess animal welfare on their farm. It was thought that the motivations of those two groups of farmers to improve animal welfare might differ too much for a fair comparison. Nevertheless, the combination of welfare and survey data provides us with valuable insights into the effectiveness of the PIGLOW app. As only descriptive statistics could be used, conservative criteria were set for when a WI was considered to have improved. Using these criteria, the data from the detailed WAs supported a clear improvement across farms for “Covered with faeces”, “Scratches”, and “Lameness”, but not for the other 16 WIs. This could indicate that some aspects of welfare are more likely to be influenced by conducting regular assessments than others. Due to the lack of a control group, it cannot be said for sure that these observed improvements were related to the use of the PIGLOW app. However, certain survey responses can shed some light on this. When asked whether they thought their farm’s performance on specific aspects of welfare had changed due to the use of the PIGLOW app, six out of eleven farmers indicated that they had noticed a positive effect on “Absence of wounds and lesions”. This was the highest number of positive scores for any of the welfare aspects. While other types of lesions were also assessed, this result does suggest that the farmers themselves noticed a change that fit with the improvement in the score for “Scratches”. For the aspects “Hygiene” (related to “Covered with faeces”) and “Lameness”, only three farmers indicated that they had noticed an improvement, despite the fact that these aspects received the lowest mean ratings for the farmers’ estimated own performance at the beginning of the study. Therefore, even though many farmers seemed to think that they could improve in these welfare aspects, they did not feel that the app helped them achieve this. A potential explanation for why the PIGLOW app might have helped farmers improve the aspect of scratches is that it requires a relatively close look to properly see the number of scratches on a pig. Because of that, taking the time to observe with the app might have stimulated farmers to think more about the causes of scratches. One farmer specifically commented on this to the researcher while conducting the first WA, saying that he noticed a large number of scratches, but that he had an idea of how to prevent them in the future. The same farmer commented in the final survey that he had started to pay more attention to lesions because of the app. This might indicate that the app had a long-term influence on how he looks at his pigs.

We can thus say that the use of the PIGLOW app likely had a positive effect on at least one and perhaps three of the WIs that were assessed. However, the fact that the other WIs did not change means that we cannot conclude that the use of the PIGLOW app had an overall positive effect on the welfare of finisher pigs on free-range or organic farms. The survey results showed that on average, the farmers noticed a medium-sized effect of the PIGLOW app on how they thought about the assessed aspects of pig welfare and a small positive effect on the welfare of their pigs. Those answers seem to fit with the data from the detailed WA. It must be noted, however, that the large standard deviations for both questions as well as comments from individual farmers showed that some farmers noticed a larger effect than others. In contrast, farmers did give very high mean scores for the PIGLOW app in general and for its user-friendliness and usefulness. Thus, it seems that the farmers were very positive about the design and format of the app, but less so about the effect it had on them and their pigs. It is worth exploring possible reasons for this.

A partial explanation for why only a small number of WIs improved could be that welfare on the participating farms was already very good at the beginning of the study. The majority of farms already had a perfect score for nine out of nineteen WIs at the start of the study. Of the individual WIs, most of which were scored between 0 and 100, no median score was higher than 4.40, indicating very good animal welfare overall. Therefore, even if the use of the PIGLOW app could potentially help improve animal welfare, it would have been difficult to detect on these farms where the potential for improvement was low. It is noticeable that the scores for the three WIs that did improve were among the highest ones at the beginning of the study. It is, therefore, possible that the higher potential for improvement explains why these were the only welfare aspects that changed. The good welfare scores on all of these farms highlight a possible issue with studies like these in which recruitment is on a voluntary basis [25]. Previous studies have shown that some farmers are uncomfortable with participating in animal welfare research, because they feel like they are being checked up on [23]. It seems likely that farmers who are proud of the welfare of their animals or their knowledge on animal welfare would feel more comfortable taking part in animal welfare studies. In addition, farmers who were motivated enough to participate in our study were likely motivated beforehand to improve the welfare of their animals in other ways. This is supported by comments made in the final survey. It also seems to align with the results of a study on pig farmers’ attitudes towards participation in animal welfare programmes [26]. That study found correlations between a high willingness to participate in animal welfare programmes and a strong belief that animals must be able to show natural behaviour, less strong objections against public discussions on animal welfare, and less negative attitudes towards animal welfare-related demands made by politicians and consumers. While this does not explicitly show that these farmers had already worked harder to improve the welfare of their animals than others, it does suggest that farmers who are willing to participate in welfare programmes have a more positive attitude towards providing good welfare for their animals. The same is likely true for the farmers in our study. If many of our farmers did not notice large benefits of using the PIGLOW app because the welfare of their pigs was already very good, they might have given high scores for the PIGLOW app itself because they did see its potential for others. This seems to be supported by some of the survey answers. One farmer gave a 9 out of 10 for the app and commented “I found the app very easy and user friendly. I think it is very suitable for a farmer who wants to improve welfare”. However, this same farmer gave a score of 2 for the influence of the app on his opinions and his pigs’ welfare, which seems to show that he was referring to hypothetical other farmers. One farmer specifically commented that he had noticed the app was very useful for young people and later added “As I said before, I think this app has value for newcomers in the sector. They can learn how to look at an animal and what to pay attention to”. Thus, the PIGLOW app could potentially help (new) farmers or others in the sector with less knowledge of animal welfare.

### 4.2. Opinions on the Welfare Assessments

While targeting a more varied group of farmers could help increase the impact of the PIGLOW app, it is also useful to take a look at farmers’ opinions on specific elements of the app to see if they can be improved. Almost all elements from the content of the WA itself received very high scores for user-friendliness as well as usefulness and were rated by (almost) all farmers. The element with the lowest score for usefulness and with ratings from only eight out of eleven farmers was “Photo material for questions”. The farmers who chose not to respond likely did not use the photo material, but as none of them commented on this, we do not know their reasons. It could be that these farmers did not need photos to familiarise themselves with certain welfare issues, but another possibility is that they were not satisfied with the available photos. In case the latter is true, it could be helpful to add additional and/or better photos of welfare problems to the app. This could be especially relevant if the app were used by less experienced farmers in the future. Looking at scores for the categories of questions, it is noticeable that “Questions on behaviours” received the highest mean score. This matches the farmers’ ranking for the effect of the PIGLOW app on the assessed aspects of welfare, which had three behavioural welfare aspects at the top (Reaction to humans, Expressing positive behaviour, Enrichment use). Thus, it seems that the PIGLOW app’s current abilities to evaluate pig behaviour were received very positively. Previous research has shown that organic farmers and farmers participating in welfare schemes often assess welfare by looking at the behaviour of pigs [27]. A general positive attitude towards assessing pig behaviour could explain why these questions were received the most positively.

### 4.3. Opinions on the Results

From all elements of the results, the usefulness of “Benchmarking” was rated by the lowest number of farmers, namely, six out of eleven. Previous studies have shown, however, that farmers often find benchmarking useful and that it stimulates many of them to make animal welfare-related changes on their farms [19,23,28,29]. Then what could explain why many of our farmers do not seem to have used PIGLOW’s benchmarking feature enough to rate it? One reason could be that the PIGLOW app was still new and that the number of users, and therefore the number of farms for which scores could be compared, was not higher than 22 at any time during the study. As the number of farms included in the comparison is shown in the app, this could have made the information less valuable to farmers. That being said, previous studies have found that providing farmers with as little as two benchmarking reports from other farms was enough to motivate them to make changes [22]. Another possibility is that some of the farmers in our study simply did not value a comparison with others as much as they valued their own data. This is supported by the fact that the highest average score for the usefulness of any element of the results was for “Keeping track of your previous data”. In the study by van Dijk et al. (2018), the majority of farmers was positive about the use of benchmarking, but at least one farmer who was quoted said that benchmarking was only useful if you thought you had welfare problems on your farm [23]. If some of the farmers in our study felt the same way, then the fact that their welfare scores were very good would indeed limit their interest in benchmarking. A final clue comes from a farmer who commented on the benchmarking feature, saying “I didn’t take the time to look for it and didn’t know at this stage that it was possible to compare ourselves to other farms. That might be the most interesting aspect”. Given that all farmers received an explanation from the researcher that visited their farm and that the results report itself contains an explanation, it is surprising that at least one farmer did not know about this element. However, the benchmarking featureis the most complex element of the results. Perhaps the explanation that is currently present in the app is not sufficient for all farmers to fully understand the benchmarking information, or it is not shown prominently enough and missed by farmers who do not read the entire report closely. Thus, it could be useful to make some changes to the explanation of benchmarking in the report or to the way in which the benchmarking data themselves are shown in order to make the feature easier to use.

### 4.4. Suggestions for Improvements

Farmers were also asked whether they had any suggestions for changes to the app. While some farmers commented that the app was good as it is, others said that they would like to see more depth and improved feedback with more practical tips and targeted solutions. Whether the remark about “depth” referred to the assessments themselves or to the feedback is not clear. The amount of depth in the assessments was based on responses to a survey on the development of the PIGLOW app that was filled out by stakeholders. In response to a question about time investment, the majority of farmers said that they would not want to spend more than one hour on an assessment. This time would be exceeded if additional elements were added or if the complexity were increased, which could lead to the loss of (potential) app users. That this is a possibility is illustrated by another survey response, which said that the app was sometimes a bit too precise. In addition, other studies have suggested that when the goal of an assessment is to gain information on the farm level, less detailed scoring systems can actually be more suitable. This is the case because they lead to a higher scoring accuracy, and the loss of precision does not hinder the achievement of the goal [30,31,32]. Adding more automated feedback, however, would not necessarily demand a larger time investment, as each farmer would only have to read the feedback for the indicators that they are interested in. The request for more practical examples of problems and their solutions suggests that, with the current feedback consisting of risk factors, farmers might have trouble identifying which of the risk factors apply to their farm. The difficulty with automated feedback in apps such as PIGLOW is that problems on farms can be related to multiple risk factors and farm-specific conditions, which makes it challenging to provide feedback that is both generally relevant and specific enough [23]. We can only try to increase the number of examples of possible causes for welfare problems in the hope that farmers can recognise which causes might be relevant for their farms. We also believe that farmers could profit from the feedback more if they followed the app’s advice to discuss their results with their veterinarians or another expert, which is also recommended by the developers of other WA tools [33,34]. This advice should perhaps be shown more prominently in the results report. In addition, as previously mentioned, this study has shown that the PIGLOW app seems most suitable for farmers who do not have as much knowledge on animal welfare yet as others. It is likely that these farmers can learn more from the feedback that is already present in the app than the average farmer in our study. Thus, besides making changes to the feedback, we can also try to promote the app among those farmers. To achieve this, we hope to be able to work together with veterinarians who can inform their clients about it. In addition, many countries have training courses for new farmers to improve their skills. In a survey organised by the European Commission among 2205 young farmers in 28 EU member states, over 50% of respondents indicated that they expected to develop skills related to improving animal welfare during such courses [35]. Thus, working together with the people in charge of those courses could be a good way to promote the PIGLOW app among young farmers who want to improve their knowledge of animal welfare.

## 5. Conclusions

The PIGLOW app did not seem to have an overall effect on the welfare of pigs on the 12 participating farms. However, the app seems to have had a positive influence on a limited number of specific welfare aspects. Farmers indicated that they found the app easy to use and that they found most of its elements very useful, but only noticed a small influence of the app on their opinions on animal welfare and the welfare of their animals. This might have been the case because the animal welfare on their farms was already very good at the beginning of the study, and so the room for improvement was relatively limited. Survey answers suggested that the farmers did see the app’s potential for helping new farmers or others who have limited experience with (monitoring) animal welfare. To promote PIGLOW among such farmers, it could be beneficial to work together with veterinarians and with people in charge of organising courses and training sessions for farmers. The farmers also offered suggestions for several aspects of the assessments and the results reports that can be improved. Reaching out to the group of potential users that could likely learn the most from the PIGLOW app and updating several elements could hopefully help a larger audience benefit from the app and further improve the welfare of pigs.

## Figures and Tables

**Figure 1 animals-14-03374-f001:**
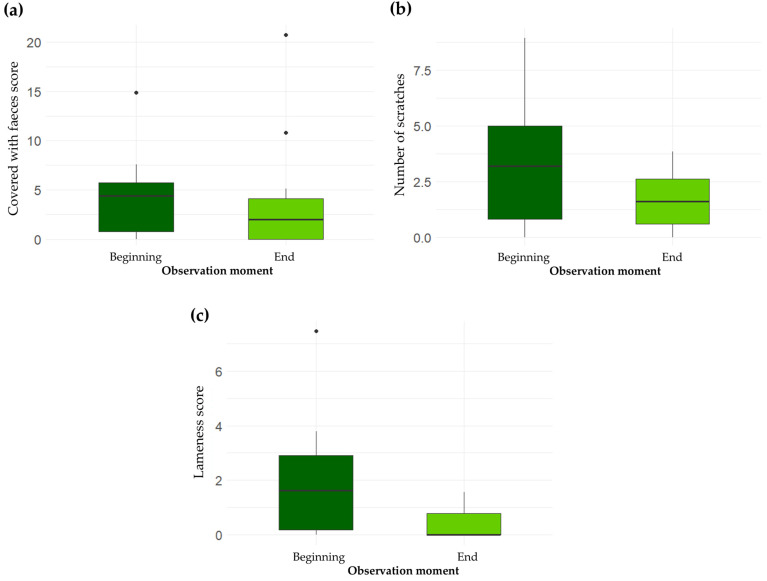
Boxplots showing the differences in the scores for the welfare indicators “Covered with faeces” (**a**), “Scratches” (**b**), and “Lameness (**c**) for finisher pigs at the beginning and end of the longitudinal study. The score for “Covered with faeces” is the mean percentage of the skin surface on one side of the body that was covered with faeces for all observed pigs. The score for “Scratches” is the mean number of scratches on one side of the body for all observed pigs. The score for “Lameness” is the mean score from 0 (best) to 100 (worst) for all observed pigs. The line in each box represents the median. *N* = 12.

**Figure 2 animals-14-03374-f002:**
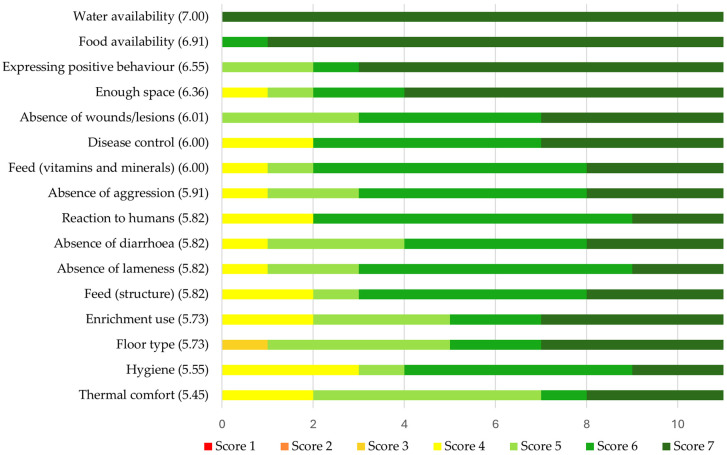
Farmers’ opinions on the importance of 16 aspects of pig welfare rated between 1 (not important at all) and 7 (very important) at the beginning of the study. All individual scores are displayed. Welfare aspects are shown in order from the highest to the lowest mean rating, which is displayed between brackets behind the name of each welfare aspect. *N =* 11.

**Figure 3 animals-14-03374-f003:**
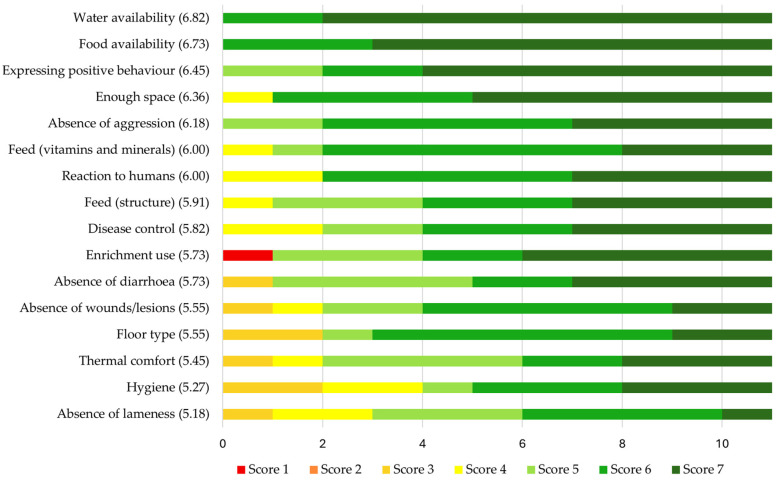
Farmers’ self-assessment of how their own farm performed on 16 aspects of pig welfare rated between 1 (very badly) and 7 (very well) at the beginning of the study. All individual scores are displayed. Welfare aspects are shown in order from the highest to the lowest mean rating, which is displayed between brackets behind the name of each welfare aspect. *N* = 11.

**Figure 4 animals-14-03374-f004:**
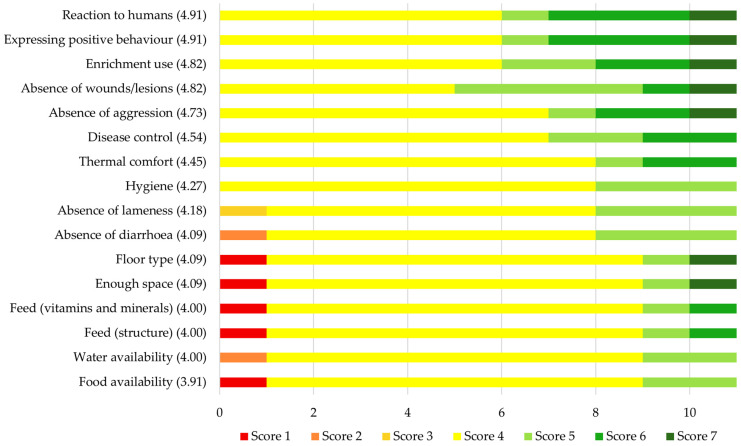
Farmers’ opinions on how the use of the PIGLOW app influenced their farm’s performance in terms of 16 aspects of pig welfare rated between 1 (severely deteriorated) and 7 (greatly improved). All individual scores are displayed. Welfare aspects are shown in order from the highest to the lowest mean rating, which is displayed between brackets behind the name of each welfare aspect. *N* = 11.

**Figure 5 animals-14-03374-f005:**
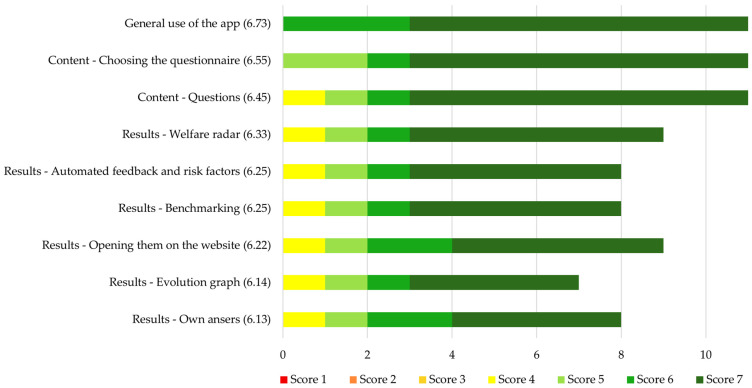
Farmers’ opinions on how easy they found it to use various elements of the PIGLOW app on a scale from 1 (very difficult) to 7 (very easy). All individual scores are displayed. The elements are shown in order from the highest to the lowest mean rating, which is displayed in brackets behind the name of each element. *N* = 11.

**Figure 6 animals-14-03374-f006:**
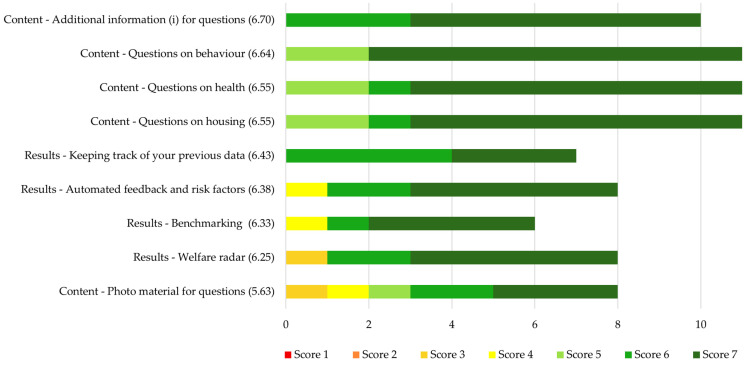
Farmers’ opinions on how useful they found each element of the PIGLOW app on a scale from 1 (not useful at all) to 7 (very useful). All individual scores are displayed. The elements are shown in order from the highest to the lowest mean rating, which is displayed in brackets behind the name of each element. *N* = 11.

**Table 1 animals-14-03374-t001:** Overview of the pig farms participating in the longitudinal study. Listed are the average number of fattening pigs on the farms during the study, whether the farms were organic or not, the type of housing system, and the dates on which the farms were visited.

Farm	Average Number of Fattening Pigs	Organic	Housing System	Date of First Farm Visit	Date of Last Farm Visit
1	808	Yes	Stable with outdoor access	27 November 2020	13 January 2023
2	200	No	Huts	7 December 2020	16 January 2023
3	385	Yes	Stable with outdoor access	22 February 2021	27 February 2023
4	600	No	Huts	10 March 2021	3 March 2023
5	21	Yes	Mobile huts	15 April 2021	9 May 2023
6	250	No	Stable with outdoor access + mobile huts	8 March 2021	17 April 2023
7	6000	Yes	Stable with outdoor access	14 January 2021	19 January 2023
8	988	Yes	Stable with outdoor access	18 January 2021	27 January 2023
9	350	No	Mobile huts	22 January 2021	23 January 2023
10	15,000	No	Stable with outdoor access	17 March 2021	1 March 2023
11	1075	Yes	Stable with outdoor access	22 February 2021	21 February 2023
12	450	Yes	Stable with outdoor access	10 August 2021	14 July 2023

**Table 2 animals-14-03374-t002:** Welfare indicators (WIs) that were assessed in the welfare assessment (WA) for finisher pigs in the PIGLOW app.

Category	Welfare Indicator	Level	Scoring Method
Good housing/environment	Covered with faeces	Individual	Number of pigs per group covered with faeces on at least 50% of the skin surface (one side of the body) is counted. Mud and sand are not counted.
	Panting	Individual	Number of pigs per group that are panting is counted.
	Shivering	Individual	Number of pigs per group that are shivering is counted.
	Lying widely spread out on the flank	Group	It is observed per group whether >50% of the pigs are spaced out across the pen and are lying on their flank.
	Huddling	Group	It is observed per group whether >50% of the pigs are huddling (lying close together and partially on top of each other).
Good feeding	Too small for their age	Individual	Number of pigs per group that are at least 1/3 smaller than the group average is counted.
	Difficulty accessing good drinking water	Group	It is observed whether any animals in the group might have difficulty accessing high-quality drinking water at any time. Access could be impeded by both physical factors (e.g., not enough drinkers) or social factors (e.g., risk of aggression).
Good health	Bad general state	Individual	Number of pigs per group that show signs of sickness or otherwise compromised health is counted. Examples: animals seem in pain, sick, needing further care to avoid complications, dull or apathic, isolated from the group, with dull/sunken eyes, blue/red ears or snout, pale skin colour, rapid respiration, or animals with a significant deformation or large hernia (bigger than the distance between the hernia and the floor).
	Lameness	Individual	Number of pigs that are lame is counted per group. Lameness is defined as everything from “visibly reduced weight bearing on one limb or limping” to being unable to walk.
	Skin wounds	Individual	Number of pigs per group that have skin wounds larger than 5 cm on the flank or legs is counted.
	Skin irritation/Parasites	Individual	Number of pigs that show signs of skin irritation (such as red spots or dots) or parasites is counted per group.
	Laboured breathing	Individual	Number of pigs that show laboured/heavy breathing (pumping) is counted per group.
	Liquid faeces	Group	It is observed per group whether there are any signs of liquid faeces inside the pen/enclosure.
	Signs of sunburn	Farm	It is noted whether signs of sunburn are observed on the farm at any time during the year.
	Coughing/Sneezing	Group	It is noted whether there was any audible coughing or sneezing at any time during the observation.
Appropriate behaviour	Use of enrichment	Individual	Number of pigs per group that are using any form of enrichment (toys, straw/roughage, or soil) is counted.
	Tail lesions	Individual	Number of pigs per group that have lesions on the tail is counted.
	Ear lesions	Individual	Number of pigs per group that have lesions on the ears is counted.
	Scratches	Individual	Number of pigs per group that have at least 15 scratches on one side of the body is counted.
	Fear of humans	Group	The observer enters the enclosure, stands still in the middle, and measures the time until they are touched by the first pig.

**Table 3 animals-14-03374-t003:** WIs that were assessed in the detailed WAs by researchers during the farm visits. The data of these WAs were used to determine the welfare status of the pigs on each farm at the beginning and end of the study. For each indicator, it is listed whether it was assessed on an individual level or at group level, whether it is positive (higher score is better) or negative (lower score is better), and how it is scored.

Welfare Indicator	Level	Positive or Negative	Scoring Method
Shivering	Individual	Negative	Yes | No. Shivering is defined as: the body shakes with (very) small movements.
Panting	Individual	Negative	Score on a continuous scale from 0 (best) to 100 (worst): 1–33 = breathing is clearly more rapid and superficial than normal; 34–66 = breaths come in gasps and chest movements are clearly more rapid than normal; 67–100 = breaths come in short gasps and chest is moving rapidly.
Laboured breathing	Individual	Negative	Score from 0 to 100: 1–25 = breathing is slightly more heavy than normal; 26–50 = breathing clearly sounds more heavy than normal, and possibly, a soft wheezing (high pitched sound) can be heard; 51–75 = more laboured breathing, possibly with more pronounced movements of the chest or a clear wheezing sound is present; 76–100 = very heavy breathing with loud wheezing or pumping and laboured movements of the chest with each breath.
Bad general state	Individual	Negative	Yes | No. Pigs in a bad general state are defined as pigs that show general signs of sickness or otherwise compromised health. Examples: animals which are obviously in pain, sick, needing further care to avoid complications, dull or apathic, isolated from the group, with dull/sunken eyes, blue/red ears or snout, pale skin colour, or rapid respiration.
Too small for their age	Individual	Negative	Yes | No. Too-small pigs are defined as pigs that are at least 1/3 smaller than the average pig in the group (of the same age).
Hernia	Individual	Negative	Score from 0 to 100: 1–25 = small protrusion and no bleeding; 26–50 = small but bleeding protrusion or moderate protrusion, but not bigger than the distance between the hernia and the floor and not bleeding; 51–75 = moderate size and bleeding hernia or bigger than the distance between the hernia and the floor but not bleeding; 76–100 = (much) larger than the distance between the hernia and the floor, up to touching the floor, and bleeding.
Skin wounds	Individual	Negative	Score from 0 to 100: 1–25 = one or several small (<2 cm), shallow wounds that are not bleeding; 26–50 = one (26) or several (50) small wounds that are open/bleeding (towards 50), one (26) or several (50) medium size (2–5 cm) wounds that are almost healed; 51–75 = a large number of small/medium and open/bleeding wounds, one large open wound or a few large (>5 cm) but almost healed wounds; 76–100 = one or more deep, large wounds that are open and/or bleeding. Wounds on all sides of the body are assessed. Wounds on the ears or tail count only as ear and tail lesions.
Scratches	Individual	Negative	The average number of scratches per side of the body (including flank, legs, and head) is counted. If both sides of the body are visible, the total number of scratches is divided by two. A mark on the skin is considered to be a scratch when it is long, relatively narrow and shallower than a wound.
Ear lesions	Individual	Negative	Score from 0 to 100: 1–25 = only very small scabs or lesions are visible; 26–50 = there are small crusts or healing wounds on the ears, but no blood; 51–75 = there are bigger crusts on the ears or smaller lesions with fresh blood; 76–100 = ears are severely damaged with big crusts and/or there are fresh wounds with blood. Scratches on the ears count as scratches, not ear lesions.
Tail lesion	Individual	Negative	Score from 0 to 100: 1–25 = only small, minor lesions are visible; 26–50 = there are slightly bigger but healing lesions, some swelling or dried blood; 51–75 = there are open wounds, significant swelling, or fresh blood; 76–100 = open wounds, significant swelling, and fresh blood.
Skin irritation	Individual	Negative	Score from 0 to 100 for the most affected side of the body: 1–25 = mild local skin inflammation or mild red spots (<10% of body surface on one side); 26–50 = larger area of mildly inflamed/spotted skin (>10%) or a small but clearly inflamed/spotted zone; 51–75 = a large area of the skin that is clearly inflamed/spotted; 76–100 = severely inflamed skin or dark spots over a large area of the skin or less severe inflammation/spots over a much larger area (>30%).
Covered with faeces	Individual	Negative	Score from 0 to 100. The score reflects the % of the skin surface of the dirtiest side of the body that is covered with faeces. Mud and sand are not counted.
Lameness	Individual	Negative	Score from 0 to 100: 1–25 = stiffness of one of the legs while walking; 26–50 = animal can walk, but weight bearing on the affected leg is clearly reduced; 51–75 = animal has clear difficulty walking; 76–100 = severe lameness that makes it (almost) impossible for the animal to walk.
Huddling	Group	Negative	Score from 0 (best) to 2 (worst). The percentage of pigs in the group that are huddling (lying partially on top of each other) is estimated/determined. 0 = No | 1 = More than 20% = | 2 = More than 50%
Enrichment use	Group	Positive	A behavioural scan is performed, and the number of pigs in the group showing object play, playing in soil/mud, or exploring their environment is counted.
Fear of humans	Group	Negative	Score from 0 to 2. After calmly entering the pen/enclosure, the percentage of pigs that respond in fear (try to get away from you, turn away from you, or huddle in a corner) is counted. 0 = <20% | 1 = 20–60% | 2 = >60%
Liquid faeces	Group	Negative	Score from 0 (best) to 3 (worst). It is assessed whether any of the faeces in the pen (on floors or walls) is liquid. 0 = None | 1 = Some | 2 = More than half | 3 = All visible faeces
Coughing	Group	Negative	Score from 0 to 3. It is noted whether there was any audible coughing at any time during the observation. 0 = No | 1 = Once | 2 = Up to 5 times | 3 = More than 5 times
Sneezing	Group	Negative	Score from 0 to 3. It is noted whether there was any audible sneezing at any time during the observation. 0 = No | 1 = Once | 2 = Up to 5 times | 3 = More than 5 times

**Table 4 animals-14-03374-t004:** Number of individual pigs and groups of pigs observed on each of the pig farms at the beginning and end of the longitudinal study.

Farm	# Individual Pigs Observed		# Groups Observed	
	Beginning	End	Beginning	End
1	30	35	6	5
2	16	19	1	3
3	47	38	14	5
4	43	37	19	5
5	17	22	2	3
6	30	40	5	6
7	54	42	19	5
8	30	32	1	2
9	30	32	5	5
10	59	33	27	6
11	15	19	1	2
12	30	36	5	2

**Table 5 animals-14-03374-t005:** Overview of all analysed questions of the two surveys for pig farmers with the corresponding answer type.

Survey	Question	Answer Type
First	In your opinion, how important is each of the following elements for good animal welfare? The pigs have enough space The right type of floor The pigs are not too cold or too hot (thermal comfort) Proper hygiene Having enough water available Having enough food available The right composition of the feed (structure) The right composition of the feed (concentration of vitamins and minerals) Disease control The absence of wounds/lesions The absence of lameness The absence of diarrhoea Possibilities to express positive behaviour Possibilities to use enrichment The absence of aggressive behaviour The reaction to human presence	Rate on an ordinal scale from 1 (not important at all) to 7 (very important)
First	Please rate how well you think your own farm performs on these same elements of animal welfare.	Rate on an ordinal scale from 1 (very badly) to 7 (very well)
Final	Overall, do you feel like the use of the PIGLOW app has changed how important these aspects of animal welfare are to you?	Rate on an ordinal scale from 1 (not at all) to 7 (very much)
Final	Please elaborate on your answer to the previous question. Which aspects of animal welfare do you find more or less important now than at the start of this study and why? Is this related to your use of the PIGLOW app?	Free text
Final	Do you feel that the use of the PIGLOW app has changed how your farm performs on these welfare indicators?	Rate on an ordinal scale from 1 severely deteriorated) to 7 (greatly improved), with 4 indicating no change
Final	Overall, do you think the use of the PIGLOW app has led to an improvement in the welfare of your animals?	Rate on an ordinal scale from 1 (not at all) to 7 (absolutely)
Final	Please add any additional remarks that you have concerning the previous question	Free text
Final	For each of the following aspects of the PIGLOW app, rate how easy you found them to understand. If you did not notice a function at all, choose “No answer”. General use of the app Content—Choosing the right questionnaire Content—Questions Results—Opening the results on the website Results—Own answers Results—Benchmarking Results—Welfare radar Results—Automated feedback and risk factors Results—Evolution graph	Rate on an ordinal scale from 1 (very difficult) to 7 (very easy)
Final	For each of these aspects of the PIGLOW app, rate how useful you found them. If you did not notice a function at all, choose “No answer”. Content—Questions on housing Content—Questions on health Content—Questions on behaviour Content—Additional information (i) for questions Content—Photo material for questions Results—Benchmarking Results—Welfare radar Results—Automated feedback and risk factors Results—Keeping track of your previous data	Rate on an ordinal scale from 1 (not useful at all) to 7 (very useful)
Final	How would you rate the PIGLOW app?	Rate on an ordinal scale from 1 (worst) to 10 (best)
Final	Please add any additional remarks that you have concerning the previous question	Free text
Final	What changes would you like to see made to the PIGLOW app?	Free text

**Table 6 animals-14-03374-t006:** Descriptive statistics for the differences in animal welfare on the pig farms at the beginning and end of the longitudinal study. The first and second columns show the medians of the mean scores per farm for each WI for the beginning and end, with the number of scores that were above 0 between brackets. The third and fourth columns show the number of farms on which the score for each indicator improved or worsened, respectively, during the study. *N* = 12.

Welfare Indicator	Median Beginning (Farms with Score > 0)	Median End (Farms with Score > 0)	Farms with Improved Scores	Farms with Worsened Scores
Too small	1.67 (6)	3.08 (8)	5	4
Bad general state	0 (4)	0 (1)	3	1
Shivering	0 (0)	0 (0)	0	0
Panting	0 (1)	0 (1)	1	1
Hernia	0 (5)	0.41 (7)	3	4
Laboured Breathing	0 (5)	0 (4)	3	3
Covered with faeces	4.40 (9)	2.01 (8)	6	4
Skin wounds	0.23 (11)	0.21 (6)	8	3
Scratches	3.20 (10)	1.60 (10)	8	2
Ear lesions	0.31 (8)	0.13 (6)	7	4
Tail lesions	0.07 (8)	0 (2)	7	2
Skin irritation	0 (3)	0 (5)	3	4
Lameness	1.63 (9)	0 (4)	8	3
Huddling	0 (4)	0 (4)	4	4
Enrichment use	20.15 (11)	19.18 (12)	5	7
Fear of humans	0.21 (6)	0.40 (7)	1	6
Liquid faeces	0 (4)	0 (3)	4	3
Coughing	0.23 (7)	0.20 (7)	5	5
Sneezing	0 (5)	0 (2)	4	2

## Data Availability

The data supporting the conclusions of this article will be made available by the authors upon request.

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
