# Peer review of "Can a Digital Application for Animal Welfare Self-Assessments by Farmers Help Improve the Welfare of Free-Range and Organic Pigs?"

_animals, 2024, doi:10.3390/ani14233374_

Round 1

Reviewer 1 Report

Comments and Suggestions for Authors

Dear Authors.

The approach of this article is very good, to use and assess the utility of a welfare assessment tool developed. To reach its full value for the readers and scientific community, the study has to be completed and the paper has to be rewritten to meet the level required for such publication. Please see my remarks and comments, meant to aid this.

Introduction

L77 – WI for the depopulation process? More specificity is needed

L79-86 – the description of the app fits more into the Materials and methods section

L106-112 – this fragment also belongs more to the Materials and methods

The aim of the study should be stated once and clearly at the end of the Introduction section, instead of suggesting it several times loosely (which leads to repetition perceived by the reader as deranging). Having segments which fit better in the Materials and methods section is not a true problem per se here, but it creates the same repetition when described in more detail in that section. Clear and concise phrasing is needed, and trust that the readers remember what they have read over a few pages.

Materials and methods

Thirteen farms assessed three and four years ago are not relevant enough for a paper of this level (small and outdated research). A solution would be to repeat the study and compare the data between, to see if farmers’ knowledge levels and attitudes changed over time. This way the number of assessed farms would increase, to enable proper statistical processing and data relevance. Additionally, the old data could be used too.

The last two columns of Table 1 seem to be switched; should not the date of last visit be more recent than the date of the first visit?

Does it have any relevance that the farms were located in two different countries? The study did not take any advantage of this situation; only relevant information should be presented and data which was used/analyzed.

How were the assessors trained, what was the inter and intra assessor agreement while performing WA and how many?

Table 2 – Title has to be in past tense. All the work performed for the study preceded its writing, to past tense has to be used throughout the paper where the work done is presented. The description of parameters is needed. What is meant by ‘good’ drinking water? Water quality standard is only one, for both humans and animals, the term ‘drinking water’ shows the standard’s level (there is no ‘good’ and ‘bad’ drinking water – i.e. the water is either proper to be consumed or not). It is unclear what is meant by ‘bad general state’, how lameness is defined (and graded) etc. etc.

Table 3 – tables and graphs have to have an informative title and content, so as these can be understood by the reader even without reading the article’s text. Revision needed in this regard. The description of parameters is needed for the detailed WA, to clarify what each indicator means.

Table 1 and L198 – no data should be presented about the farmers/farms which were excluded from the study, even if at the end of it. Irrelevant information and it is misleading for the reader. Especially in L198, it is not clear which farmer/farm was excluded. The paper can mention that the study started with 13 farms/farmers but only 11 were included in the end, can mention the exclusion causes, but the excluded participants’ data/information has no place in the article.

Table 5 – why is only this questionnaire given in the article? Either provide the questionnaires and assessment protocols in full as annexes of the paper, or describe them briefly but in enough detail to clarify each parameter/item/question.

L238-240 – for statistical analysis a different approach regarding the qualitative data might have been preferable. How many farmers had the same opinion? This is not clear enough for the reader; we only know that each opinion was selected to be shown.

Results

L254-258 – if there was a difference (even if only of one digit), was it statistically significant? This kind of phrasing should be used here, if the data were statistically processed. 

Table 6 – the results of the excluded farm should not be presented!

 There is no statistical significance given, although it was stated that comparisons were performed, thus the results have no statistical relevance.

Although it was stated to be performed, the statistical analysis is not presented.

Discussion

The Covid-19 pandemic should not be used as an excuse for this study. The time passed is already a disadvantage needing a solution (e.g. performing a new batch of assessments for comparison with the old data).

L452 – very weak argumentation. If the researchers are not convinced of the value of their work, the readers cannot be.

Not enough references are used, to position the results in the existing knowledge pool – and this should be the role of the Discussions section. A major part of this section presents again the results and this is not a proper approach. This section is repetitive and contains irrelevant data – a major problem of the article to be solved.

Conclusions

Repeats too much the information already stated in other sections, repeats the results and not focused enough to conclude (also too long).

Author Response

Thank you for your feedback. Please find our reply to all of your comments in the attached file.

Reviewer 2 Report

Comments and Suggestions for Authors

This is a very small pilot study with only 11 farmers completing both the assessments and the surveys. The farms do not appear to be representative, and no information is provided on how they were selected. Although welfare indicators included huddling, shivering, panting and lying spread out, temperatures or other weather conditions at the time of assessment were not recorded.

I strongly recommend including data from the initial survey to strengthen this manuscript. Quality above quantity.

Detailed comments are in the attached file.

Author Response

(The authors gave the same response as above.)

Reviewer 3 Report

Comments and Suggestions for Authors

This manuscript is very well written and easy to read. Despite the (regrettable) low sample size, I believe that this study is interesting for the scientific community as well as for the industry. 

I am suprised that the authors did not consider doing non-parametric statistical tests on their data, but I also acknowledge that descriptive statistics are the most "honest" way to present the results of this study (P-values would not add much to the conversation).

In order to be completely transparent about the main limiting factor of the study, I recommend to state the sample size in the abstract and in the conclusion of the manuscript. I also recommend to add in the abstract that descriptive statistics were used to present the results. Stating in the title of the manuscript that this is an exploratory study, or a proof of concept might also be considered.

Finally, I have a couple of minor comments:

Table 1 : the dates of first and last farm visits are inverted.

Farm number 10 was visited on two consecutive days, not 2 years apart

L130: I need more details on how the assessments were performed: how were the assessed groups selected (i.e. representative percentage of the farm, fixed number, random...) ? How were individual parameters scored (e.g. all pigs in a given pen, or randomly selected pigs)?  If you did not control that, it also needs to be stipulated.

L152-153: Thanks for the precision, but what explains the difference in number of groups observed between the first and last visits (i.e. farms 2, 3, 4, 7 , 10 in table 4)

Table 3: 

It is not very clear what "positive" or "negative" stand for here - I guess it means that the higher the score, the more positive/negative it is for the welfare ? I advise to specifies that in the caption or as a footnote.

I would suggest using the same terminology as in Table 2, so here, "too small for their age".

Maybe group items that belong together (e.g. laboured breathing and panting, all skins-related assessments)

I think that a definition of "skin wounds" and "scratches" would be great, as these two can be easily confounded.

Table 4: I suggest numbering the farms as in Table 1 (just delete 7 and number 1-6 and 8-13) - that allows for easier comparisons.

L304: "those" instead of "other farmers"

Figures 2 and 4: it is a little difficult to distinguish between score 2 and score 3. The distinction is better in Figure 2 as the two scores are represented, but in Figure 4 only one is represented (score 3?).

L418-423: Fair point about the control group, I appreciate that the authors detail this limitation and thus provide guidance for future work.

L435-439: But the farmers were aware of their scores before they answered the survey right ? Therefore, they could have been biased by their own results.

L536-538: were the 11 farmers the only users of the app? If so, make it clear, if not, state the number of users.

L647: Typo in the name of Valérie Courboulay.

L689-690 - Ref 16: maybe add the URL: https://pureportal.ilvo.be/en/publications/piglow-a-welfare-self-assessment-amp-benchmarking-tool-for-outdoo

Author Response

(The authors gave the same response as above.)
